# SwinBTS: A Method for 3D Multimodal Brain Tumor Segmentation Using Swin Transformer

**DOI:** 10.3390/brainsci12060797

**Published:** 2022-06-17

**Authors:** Yun Jiang, Yuan Zhang, Xin Lin, Jinkun Dong, Tongtong Cheng, Jing Liang

**Affiliations:** College of Computer Science and Engineering, Northwest Normal University, Lanzhou 730070, China; jiangyun@nwnu.edu.cn (Y.J.); 2020222054@nwnu.edu.cn (X.L.); 2020222078@nwnu.edu.cn (J.D.); 2020211957@nwnu.edu.cn (T.C.); 2020211969@nwnu.edu.cn (J.L.)

**Keywords:** brain tumor segmentation, Swin Transformer, 3D CNN, depth-wise separable convolution

## Abstract

Brain tumor semantic segmentation is a critical medical image processing work, which aids clinicians in diagnosing patients and determining the extent of lesions. Convolutional neural networks (CNNs) have demonstrated exceptional performance in computer vision tasks in recent years. For 3D medical image tasks, deep convolutional neural networks based on an encoder–decoder structure and skip-connection have been frequently used. However, CNNs have the drawback of being unable to learn global and remote semantic information well. On the other hand, the transformer has recently found success in natural language processing and computer vision as a result of its usage of a self-attention mechanism for global information modeling. For demanding prediction tasks, such as 3D medical picture segmentation, local and global characteristics are critical. We propose SwinBTS, a new 3D medical picture segmentation approach, which combines a transformer, convolutional neural network, and encoder–decoder structure to define the 3D brain tumor semantic segmentation job as a sequence-to-sequence prediction challenge in this research. To extract contextual data, the 3D Swin Transformer is utilized as the network’s encoder and decoder, and convolutional operations are employed for upsampling and downsampling. Finally, we achieve segmentation results using an improved Transformer module that we built for increasing detail feature extraction. Extensive experimental results on the BraTS 2019, BraTS 2020, and BraTS 2021 datasets reveal that SwinBTS outperforms state-of-the-art 3D algorithms for brain tumor segmentation on 3D MRI scanned images.

## 1. Introduction

Brain tumors pose a serious threat to human life. Currently, there are more than 100 types of brain tumors affecting humans [1]. The treatment methods for such diseases include surgery, chemotherapy, and radiotherapy. With the continuous development of artificial intelligence, tumor diagnosis and surgical pre-assessment interventions based on artificial intelligence are playing an increasingly important role. Fine segmentation of brain tumors using techniques, such as voxel analysis, can help one to study their progression and assist in preoperative planning [2]. Brain tumor segmentation from brain tumor images is currently at the forefront of research [3,4,5].

Magnetic resonance imaging (MRI) technology can provide images of different contrasts (i.e., modalities) and is a non-invasive, high-performance soft tissue contrast imaging modality [6]. A complete MRI image includes four modalities: T1-weighted (T1), T1-enhanced contrast (T1-ce), T2-weighted (T2), and T2 fluid-attenuated inversion recovery (Flair). Each of the four modalities captures specific features of the underlying anatomical information. Combining multiple modalities can provide highly comprehensive information for analyzing different subregions of organs and lesions. Among them, T2 and Flair images are suitable for detecting edema around the lesion. T1 and T1ce are suitable for detecting the core of the lesion. Generally speaking, there are obvious differences in the gray level of the lesion area and normal tissue in Flair images, while the boundary features of the lesion area in T1ce images are more obvious [3,4,5]. MRI technology can produce high-quality brain images without damage and skull artifacts and can provide more comprehensive information for the diagnosis and treatment of brain tumors, including the shape, size, and location of organs and lesions. It plays a key role in diagnosis and is the main technical means of brain tumor diagnosis and treatment.

For the auxiliary diagnosis technology of medical images, some studies focus on the fusion of images. For example, CSID [7] proposes an algorithm for fusing CT and MRI images to enhance the detailed information of clinical diagnosis, but the most important research at present is image segmentation, such as using modified-moth-flame algorithm and Kapur`s thresholding for evaluating brain tumor [8], or using deep learning methods to segment brain tumor images. 

In recent years, deep convolutional neural networks, such as Alex-Net [9], VGG-Net [10], ResNet [11], and Google-Net [12], have been successfully applied to many computer vision tasks, and have been maintaining SOTA performance. Due to the powerful feature extraction capability of deep convolutional neural networks, they were soon applied to the field of medical image processing and analysis [13,14,15]. In brain tumor segmentation, the method using fuzzy edge detection and U-NET CNN classification [16] can exceed the performance of traditional machine learning methods; convolutional-neural-network-based segmentation methods have also achieved state-of-the-art performance in various tests [17,18,19,20]. However, due to the small and fixed size of the convolutional kernel of CNNs, although the dilated Convolution [21,22] expands the perceptual field and the deformable Convolution [23] allows some offset in the kernel, they find it difficult to extend adaptively and flexibly to the entire feature map. This limits their ability to learn global features and remote features, which are crucial for the accurate segmentation of tumors of various shapes and sizes.

Transformer models have excellent performance in natural language processing tasks [24,25]. In the last two years, ViT [26], an approach that introduces a transformer to computer vision tasks, has also achieved state-of-the-art performance in that time. ViT relies on the global and remote modeling capabilities of a transformer and can achieve better performance than CNNs, but the huge number of parameters in this class of models makes it easy to implement in 2D segmentation tasks, while it is subject to the memory limitation of the graphics card in 3D segmentation tasks. For example, the U-Netr method [27] designed a network model using ViT as an encoder. Although this method showed better performance, the number of model parameters reached more than 100 M. The training of the model was time consuming and labor intensive. Swin Transformer [28] proposed a hierarchical visual transformer using a shifted window, which restricts the self-attention to the window. The performance of the SwinUnet [29] method based on the Swin Transformer method is also powerful. The reduction in the number of parameters makes it easier to introduce the transformer into the 3D segmentation task. In this work, we propose a novel method called SwinBTS, which uses an encoder–decoder architecture, utilizing the 3D Swin Transformer module as an encoder to extract contextual information and connect to a decoder with the same resolution by skip-connection. The decoder also uses the 3D Swin Transformer module. The NFCE (Neighbor-Feature Connection Enhancement) module is used between the encoder and the downsampling to enhance the feature information between the transformer structure and the convolutional downsampling with a step size. The NFCE module is also added between the decoder and the upsampling. The resulting method is found to be insufficient for detailed feature extraction after experiments. For the extraction of local detail features, effective methods are channel attention, spatial attention, convolution, etc. After comparing the experiments and the inspiration provided by the method ELSA [30], we propose a module ETrans (Enhanced Transformer) in the bottom BottleNeck part of the model combined with the matrix operation of Hadamard product, which has the structure of a transformer and is mainly implemented with a convolution operation to extract local information.

Experiments on the BraTS 2019, BraTS 2020, and BraTS 2021 datasets demonstrate the effectiveness of our model.

The main contributions of this work are as follows:We propose a new transformer-based method for 3D medical image segmentation.In this method, we designed a combination of Transformer structure and CNN to achieve better performance, and we also designed a module ETrans to enhance detail feature extraction.The proposed model achieves excellent segmentation results on BraTS 2019, BraTS 2020, and BraTS 2021 datasets.

The structure of this paper is as follows. Section 2 details the 3D medical image segmentation method using CNN and the 3D medical image segmentation method using a transformer. Section 3 mainly introduces the overall network structure and its various components. Section 4 introduces the experimental dataset, experimental environment, and experimental hyperparameters. We compare and analyze the experimental results. Finally, a summary is presented in Section 5.

## 2. Related Work

### 2.1. Convolution-Based 3D Medical Image Segmentation Method

Applying deep convolutional neural networks to 3D image segmentation, the first proposed method is 3D U-Net [31], which is a 3D version of U-Net [32] and can achieve good segmentation results using an encoder–decoder structure. Later, V-Net [33] proposed the loss function Dice Loss for the first time, and the training period was optimized according to the Dice coefficients to achieve a state that can handle the existence of a strong imbalance between foreground and background voxels. nnU-Net [34] did not innovate the network structure. It is a powerful segmentation model focused on dataset preprocessing, model training methods, inference strategies, etc. Many of the current BraTS challenges use this method as a baseline model for innovation. AGU-Net [17] is proposed to add an attention gate mechanism to ResU-Net and achieved good results. ERV-Net [35] improved 3D U-Net by introducing residual blocks and also adding the lightweight network ShuffleNetv2 [36] to the encoder to reduce the complexity of the model, which achieved the state-of-the-art performance on the BraTS 2018 dataset.

### 2.2. Transformer-Based 3D Medical Image Segmentation Method

The current research mainly combines a transformer and CNN to achieve the purpose that the network model can extract both global information and local information. nnFormer [37] uses a transformer as an encoder and decoder and uses convolution as downsampling and upsampling. TransBTS [38] is the first attempt to utilize transformers for 3D multimodal brain tumor segmentation by efficiently modeling local and global features in both spatial and depth dimensions. Specifically, TransBTS`s encoder–decoder architecture uses a 3D CNN to extract local 3D volumetric spatial features and a transformer to encode global features. The method proposed is more efficient than CNN-based methods. TransBTSv2 [39] improves the model segmentation, mainly by introducing deformable convolution in the skip-connection part based on TransBTS. BiTr-UNet [40] differs from TransBTS in that the model adds two ViT layers in the deep skip-connection part to model global features. Unetr [27], on the other hand, used ViT layers as encoders and convolutional layers as decoders to build the network. The method achieved excellent performance on several tasks, but the model resulted in a large number of parameters due to a large number of ViT layers used. VT-Unet [41] is a lightweight model for segmenting 3D medical images in a hierarchical manner. It introduces two self-attention layers in the encoder to capture local and global information. This model also introduces window-based self-attention, cross-attention modules, and Fourier position coding in the decoder part to significantly improve accuracy and efficiency. Cotr [42] designed a deformable transformer encoder, which focuses on only a small portion of the key location feature information, which also greatly reduces the computational complexity and spatial complexity. the experimental results show that this method has a significant improvement in effectiveness compared to other transformer and CNN combination methods. TransFuse [43] combines a transformer with CNN in parallel to efficiently capture global dependencies at a shallow level of the network. MissFormer [44] redesigned the feedforward network using an augmented transformer to enhance remote dependencies and complement contextual information.

There are key differences between our model and these efforts: SwinBTS uses a Swin Transformer as the encoder and decoder rather than as an attention layer.SwinBTS combines a transformer and CNN to form the entire network, combining the advantages of both.SwinBTS designs an enhanced transformer module for cases where detailed features are under-extracted.

## 3. Methodology

### 3.1. Overall Architecture

The overall architecture of the proposed SwinBTS is shown in Figure 1. Concretely, given a multimodal MRI medical image input X∈ℝC×H×W×D, where the image space size is H×W×D, the number of channels (modal number) is C. Because of the excellent performance of the encoder–decoder structure in the segmentation task, we use the encoder–decoder structure for the overall framework design. The 3D Swin Transformer module and downsampling are first used to extract spatial and semantic information to deepen the network depth, respectively, and then the enhanced transformer module is used to extract deeply detailed feature information. The upsampling and 3D Swin Transformer module are used to gradually produce segmentation results with the same resolution as the input. Next, we will describe the individual components of SwinBTS in detail.

### 3.2. Network Encoder

#### 3.2.1. 3D Patch Partition

For the encoder, we cut the medical image into non-overlapping patches using a 3D Patch Partition layer to convert the input to a serial input (patch size is 4×4×4). After cutting into patches, we used linear embedding layers to map the patches to *C*(4×4×4=96) dimensional vectors. With this division, we obtain a feature map of size C×(H/4)×(W/4)×(D/4).

#### 3.2.2. 3D Swin Transformer Module

An important reason limiting the application of transformer structure to medical image tasks is that converting the whole image into a sequence for self-attention is too computationally intensive, consumes too much memory, and consumes a lot of time for training the model. In contrast, Swin Transformer is built based on a shifted window, which greatly reduces the number of parameters and at the same time exhibits a much better feature learning capability. The structure of the Swin Transformer block is shown in Figure 1 for the 3D Swin Block, which consists of LayerNorm (LN) layer, window-based multi-head self-attentive module, GELU, and multilayer perceptron (MLP). In this paper, we extend the Swin Transformer block to 3D as the base module of the whole network.

#### 3.2.3. Downsampling Module

Downsampling is implemented using a convolution operation with strides rather than a neighboring concatenation operation. The method nnFormer [37] shows that convolution downsampling improves the performance of the model because convolution downsampling produces a hierarchical representation which helps to model the object concept at multiple scales. The specific implementation is to use a convolutional with kernel size 2×2×2, and stride is 2. After that, the LayerNorm function is used for normalization, and finally, the GELU function is used for activation. We add the Neighbor-Feature Connection Enhancement (NFCE) module before the downsampling module to enhance the feature information between the 3D Swin Transformer module and the downsampling module to reduce the information loss. The NFCE module is a deep separable convolution with residual structure, as shown in the NFCE Block in Figure 1.

The whole encoder is implemented by stacking the 3D Swin Transformer module with the downsampling module, and after three downsamplings, the image size changes from (H/4)×(W/4)×(D/4) to (H/32)×(W/32)×(D/32), and the number of channels is also increased by a factor of 8, enabling feature extraction of the input image.

### 3.3. Network Decoder

The decoder part also uses the 3D Swin Transformer module for feature decoding, with the same depth as the encoder. A skip connection is used between the encoder and the decoder. The upsampling module is performed using deconvolution. The same NFCE module is used between the 3D Swin Transformer module and the upsampling module to enhance the feature information. After the last layer of the 3D Swin Transformer module, we obtain the feature map with a size of C×(H/4)×(W/4)×(D/4). To achieve pixel-level prediction, it is necessary to obtain H×W×D resolution output, which we achieve by enhancing the dimensionality of the feature maps through linear transformations and then assigning them to image sizes. A brain tumor is a triple classification task, and the final output will be the feature map with size 3×H×W×D.

### 3.4. Enhanced Transformer Module

To enhance the model for detailed feature extraction, an ETrans (Enhanced Transformer) module is added between the underlying encoder and decoder (BottleNeck), as shown in Figure 2. Combining the methods [30], a common view in deep learning is that higher-order mappings have a stronger fitting capability. Both the attention mechanism as well as the convolution are second-order mappings. The structure of the attention mechanism, as an example, is shown in Equation (1):(1)$$y_{i}=Soft\;\max(f(x_{i}))x_{i}+x_{i}$$
where f(·) denotes a series of convolution operations, $x_{i}$ is the input feature map, and $y_{i}$ is the output feature map. From Equation (1), we can see that the attention structure is a second-order mapping, while the self-attention structure in the Transformer structure is a third-order mapping, which may be the reason why the final result is not improved. However, how do design the convolutional attention as a third-order mapping? We implemented this structure by applying the Hadamard product [45], which is chosen because it is computationally simple and consumes less memory compared to the matrix product. Further, in combination with the Transformer structure, the same MLP module is added after the attention structure to obtain our structure, as shown in Equations (2) and (3):(2)$${\hat{y}}_{i}=Soft\;\max(f(H\_k⊙H\_q))H\_v+x_{i}$$
(3)$$y_{i}=MLP(LN({\hat{y}}_{i}))+{\hat{y}}_{i}$$
where f(·) is also the convolution operation, ⊙ denotes the Hadamard product, H_k, H_q and H_v are the feature maps obtained from the input linear transform, respectively. We also experimentally verify the effect of stacking this module.

## 4. Experiments

### 4.1. Datasets

The experiments were conducted mainly on three public multimodal brain tumor datasets BraTS2019, BraTS2020, and BraTS2021 [3,4,5,46,47]. All three datasets are competition data provided by the BraTS challenge, which aims to evaluate state-of-the-art methods for semantic segmentation of brain tumors by providing 3D MRI datasets with Ground Truth annotated by physicians. BraTS 2019 contains 335 cases of brain images for training, with each sample consisting of four brain MRI scans, namely T1-weighted (T1), T1-enhanced contrast (T1-ce), T2-weighted (T2), and T2 fluid-attenuated inversion recovery (Flair). The volume of each mode is 240×240×155, which has been aligned into the same space. The labels contain four categories: background (Label 0), necrotic and non-enhancing tumors (Label 1), peritumoral edema (Label 2), and GB-enhancing tumors (Label 4), for segmentation of the enhanced tumor region (ET, Label 4), the core tumor region (TC, Labels 1, 4), and the entire tumor region (WT, Labels 1, 2, 4). The BraTS 2020 dataset contains 369 cases of training data and 125 cases of validation data (unlabeled, for online validation), the BraTS 2021 dataset contains 1251 cases of training data and 219 cases of validation data (unlabeled, for online validation). Except for the number of cases in the dataset, all other data of BraTS 2020, BraTS 2021, and BraTS 2019 are the same.

### 4.2. Implementation Details

The model parameters are initialized using the weights pre-trained by Swin-T on ImageNet-1K. For training, we use the Adam optimizer to train the model with an initial learning rate of 1 × 10^−4^ with a cosine decay strategy. The source code can be found at https://github.com/langwangdezhexue/Swin_BTS (accessed on 18 May 2022). The following data enhancement techniques are used:Min-max scaling followed by clipping intensity values;Crop the volume to fixed size 240×240×155 by removing unnecessary backgrounds.

Our loss function is a combination of dice loss and cross-entropy loss, and it can be computed in a voxel-wise manner according to Equations (4)–(6):(4)$${ℒ}_{dl}(G,Y)=1-\frac{2}{J}\sum_{j=1}^{J}\frac{\sum_{i=1}^{I}G_{i,j}Y_{i,j}}{\sum_{i=1}^{I}G_{i,j}^{2}+\sum_{i=1}^{I}Y_{i,j}^{2}}$$
(5)$${ℒ}_{ce}(G,Y)=\frac{1}{I}\sum_{i=1}^{I}\sum_{j=1}^{J}G_{i,j}\log Y_{i,j}$$
(6)$$ℒ(G,Y)={ℒ}_{dl}(G,Y)+{ℒ}_{ce}(G,Y)$$
where I is the number of voxels, J is the number of classes, $Y_{i,j}$ and $G_{i,j}$ denote the probability output and one-hot encoded ground truth for class j at voxel i, respectively.

In the model, we use the Dropout operation, which is to make each neuron in a state of inactivation with a certain probability in the forward propagation of the training process to achieve the purpose of reducing overfitting.

### 4.3. Evaluation Metrics

We use the Dice score and 95% Hausdorff Distance (HD) to evaluate the accuracy of segmentation in our experiments. The Dice score and HD metrics are defined as:(7)$$\mathrm{Dice}(G,P)=\frac{2\sum_{i=1}^{I}G_{i}P_{i}}{\sum_{i=1}^{I}G_{i}{}_{i}+\sum_{i=1}^{I}P_{i}}$$
(8)$$\mathrm{HD}(G^{\prime },P^{\prime })=\max\{\underset{g^{\prime }\in G^{\prime }}{\max}\underset{p^{\prime }\in P^{\prime }}{\min}||g^{\prime }-p^{\prime }||,\underset{p^{\prime }\in P^{\prime }}{\max}\underset{g^{\prime }\in G^{\prime }}{\min}||p^{\prime }-g^{\prime }||\}\;$$

For a given semantic class, let $G_{i}$ and $P_{i}$ denote the ground truth and prediction values for voxel i, $G^{\prime }$ and $P^{\prime }$ denote ground truth and prediction surface point sets, respectively. The 95% HD uses the 95th percentile of the distances between ground truth and prediction surface point sets. As a result, the impact of a very small subset of outliers is minimized when calculating HD.

### 4.4. Experiment Results

#### 4.4.1. BraTS 2019 Dataset

On this dataset, we mainly perform model validation, and we divide 335 samples into 222, 57, and 56 cases as training, validation, and test sets, respectively. The Dice scores and the average Dice scores of SwinBTS on this dataset for ET, TC, and WT categories reach 74.43%, 79.28%, 89.75%, and 81.15%, respectively. We trained some SOTA models with the same dataset partitioning for use as a comparison, and the experimental data are shown in Table 1.

In general, SwinBTS has good performance in ET, TC, and WT categories. Table 1 shows that compared with the classical 3D U-Net, the Dice score of SwinBTS has a great advantage, which shows the advantage of the Transformer structure compared with convolution. The results show that the method can have stronger segmentation performance than 3D U-Net. U-Netr uses the transformer structure as the encoder, which is more conducive to learning long-distance features, so the effect of TC and WT category segmentation is more obvious. TransBTS adds a Transformer structure to 3D U-Net to model global relationships, but the method does not perform well in the ET category, indicating that the extraction of local features is insufficient. VTU-Net is a method established using the transformer structure. The Dice scores of each category are well balanced while SwinBTS has an excellent performance in the ET and TC categories, indicating the excellent extraction ability of our method for global features and local features.

The data in brackets in Table 1 is the standard deviation of the segmentation results, from which it can be seen that the SwinBTS model has the smallest classification variance for the three categories, indicating that the model has the best segmentation effect and the most stable segmentation, and the segmentation results will not have large differences.

In Table 2 we compare the mIOU results on the 2019 test dataset, and we also compare the standard deviation. Compared with 3D Unet, SwinBTS can achieve a 10.07% higher mIOU result, and also has a 1.03% improvement compared to the VTU-Net model that also uses the Transformer structure. The smaller standard deviation also shows the stability of the SwinBTS model segmentation results.

We also draw the boxplots of the Dice scores of the SwinBTS and TransBTS methods on the BraTS 2019 test dataset for comparative analysis, as shown in Figure 3. These two methods achieve high-performance segmentation in most test samples. However, since the dataset itself is obtained in different ways, the data are easily disturbed by factors, such as noise, so there will be outliers, resulting in lower segmentation results.

#### 4.4.2. BraTS 2020 Dataset

This dataset is mainly used for comparison with the SOTA model. We divide the training set in the dataset into a training set and a validation set for training at a ratio of 8:2 and then perform segmentation prediction on the BraTS 2020 Validation dataset. The results were submitted to the BraTS 2020 Challenge official website for online verification and comparison with the SOTA model. The dataset evaluates the results according to the main evaluation indicators of the challenge, Dice score, and 95% Hausdorff distance. The results are shown in Table 3.

The Dice scores of the SwinBTS method in the three categories of ET, TC, and WT are 77.36%, 80.30%, and 89.06%, respectively. The Hausdorff distances are 26.84 mm, 15.78 mm, and 8.56 mm, respectively. Compared with traditional CNN methods, such as 3D U-Net, V-Net, and Residual U-Net, SwinBTS has obvious improvement. It also has a certain improvement compared with methods using transformer structures, such as U-Netr, TransBTS, and VTU-Net. We can also see that the improvement in the SwinBTS model in Table 3 is relatively limited compared to the VTU-Net model, so the standard deviation of the Dice score is compared, and it is found that the standard deviation of the SwinBTS model is much lower, indicating that the model is in a large number of segmentation tasks. The model is much more stable and does not exhibit large deviations.

#### 4.4.3. BraTS 2021 Dataset

This dataset has the same settings as the BraTS 2020 dataset. It also uses the 8:2 ratio split dataset as the training set and the verification set for training, and finally, conducts online verification. The results are shown in Table 4.

SwinBTS also achieved excellent segmentation results on the BraTS 2021 dataset. ET, TC, and WT can achieve Dice scores of 83.21%, 84.75%, and 91.83%, respectively, and the Hausdorff distances are 16.03 mm, 14.51 mm, and 3.65 mm, respectively, exceeding the results of most methods.

### 4.5. Ablation Experiments and Analysis

We also conducted sufficient ablation experiments to verify the effectiveness of the SwinBTS model. There are mainly two ablation experiments:

(1) We verify the validity of each module, as shown in Table 5. The segmentation results of the basic model, adding the NFCE module, adding the Transformer module to the Bottleneck model, adding the convolution module to the BottleNeck model, and adding the ETrans module, are compared.

We first convert SwinUnet to a 3D version as the basic method of the study. From the experimental results, we can see that the average Dice score obtained by SwinUnet3D is 78.96%, which is lower than that of TransBTS and other methods. After adding the NFCE module, the Dice score can be improved by 0.87%. This module improves all three categories’ results. In the bottom bottleneck part, we first tried to add a Transformer structure and a convolution module, respectively. The results show that adding the Transformer structure can improve the performance to a certain extent, but compared with the SOTA model, the model has poor performance for ET and TC category segmentation. For these two categories, information extraction requires the model to have strong detail feature extraction capabilities. Therefore, we chose to use convolution to enhance the model, but after adding the convolution module, the model performance dropped by 0.37%. The reason for the analysis may be that the convolution structure is a second-order mapping, resulting in insufficient model fitting ability. Therefore, we combine the Transformer structure to transform the convolution operation into a third-order map and add the MLP structure to design the ETrans module. The final experiment also proves the effectiveness of this module. The average Dice score is 1.32% higher than that without this module and 2.19% higher than the baseline model.

(2) Depth ablation experiment with the ETrans module. The segmentation outcomes when the number of stacks in this module is 1, 2, or 4, respectively, are shown in Table 6.

From Table 5, we can see that it is not the case that the higher the number of stacks, the better the segmentation effect is. We follow the usual setting that the segmentation result when the depth is four is not as good as the segmentation result when the depth is two. Therefore, the depth of ETrans used in this paper is two.

### 4.6. Heatmap Analysis

Figure 4 shows the heatmap of the model before and after adding the ETrans module. The role of the ETrans module is mainly to improve the model’s feature extraction capability for local features, especially small-size categories. From Figure 4, we can see that when the ETrans module is not added, the model has poor recognition of the necrotic area and the enhanced tumor area in the entire tumor area, so the segmentation ability of the two categories of ET and TC is poor. After adding the ETrans module, the network obviously pays more attention to the central area of the tumor.

### 4.7. The Impact of Dataset Noise on Experiments

Noise is an unavoidable problem in medical images. Due to different factors, such as equipment, operations, patients, and environments, datasets always have different levels of noise problems. Therefore, in this section, we explore the segmentation ability of the SwinBTS model for datasets with varying degrees of noise.

Our main approach is to add different degrees of Gaussian noise to the BraTS2019 dataset. The dataset comparison after adding noise is shown in Figure 5.

In Table 7, we enumerate the effect of adding different degrees of noise on the segmentation effect.

From Table 7, we can see that SwinBTS is more sensitive to noise. When the noise is low, the model still has excellent performance, but when the added noise level (noise-sigma = 5) is large, the final segmentation result of the model will drop by about 10%. Therefore, in the task of medical image analysis, noise is a key influencing factor, but for MRI images, the noise factor has less influence, and it does not have a great impact on brain tumor segmentation tasks.

### 4.8. Visual Comparison

In this section, we compare the visualization of the brain tumor segmentation results between the proposed method and 3DU-Net, TransBTS, and VTU-Net, as shown in Figure 4. In Figure 6, the first row is the cross-sectional image of the brain tumor, the second row is the sagittal image, and the third row is the coronal image. For the convenience of observation, we show all three cross-sectional images and set the coordinates of the intercept point as (109, 89, 78). The red label in the figure represents the ET area, the yellow label represents the TC area, and the green label represents the WT area. It can be seen from Figure 6 that all models have the best segmentation effect for the WT region, and the segmentation effect for the two complex edges of ET and TC is very different. Compared with Ground Truth, our model is more accurate for the segmentation results of edge details.

### 4.9. Discussion

Using artificial intelligence to assist doctors in diagnosis can greatly improve the efficiency of diagnosis. The use of deep learning methods for medical image segmentation is currently the most cutting-edge research and has the best performance. However, in order to truly apply this research to medical-aided diagnosis, higher segmentation accuracy is required to ensure the safety and effectiveness of diagnosis. The main purpose of our research is to improve the segmentation accuracy of brain tumor MRI images by designing and improving the network model. The comparison of the above experimental results also proves that our model has excellent segmentation performance (Dice score). However, our model also shows some inadequacies; it can be seen from Table 3 that the SwinBTS model performs worse than the Unetr model and the TransBTS model in the 95% Hausdorff Distance indicator, indicating that the SwinBTS model has relatively insufficient ability to segment image edges. We think this problem is caused by the extensive use of the Transformer structure. Therefore, our next step will be to combine CNN on the basis of SwinBTS to improve it and achieve better performance.

## 5. Conclusions

We proposed a novel segmentation method, SwinBTS, which can automatically segment diseased tissue in brain MRI images. The model effectively combines a 3D Swin Transformer, 3D convolutional structure, and encoder–decoder structure to achieve efficient and accurate segmentation of MRI images. Different from using CNN, we use a 3D Swin Transformer as the encoder and decoder to effectively extract the global information of the feature map. After the encoder/decoder, the combination of NFCE module and downsampling/upsampling can reduce information loss when downsampling/upsampling. The ETrans module added at the bottom of the model is designed by combining CNN and Transformer structures. This module is used to extract local detailed features, so that the model also has strong segmentation capabilities for categories that occupy a small proportion of the image (such as ET). Finally, we validate the method on three datasets (BraTS 2019, BraTS 2020, and BraTS 2021). Experimental results show that our method has better performance in brain tumor MRI image segmentation compared with some state-of-the-art methods (such as Residual U-Net, Attention U-Net, and TransBTS). Multiple experimental results show that our method achieves good results on three datasets, indicating its potential for practical application in auxiliary diagnostic systems. The visualization results show that the proposed method has good segmentation performance for all three lesion regions of brain tumors. In future work, we will explore optimizing the self-attention structure of the Swin Transformer module to improve the overall performance of the method, and at the same time, explore how to more effectively combine the Transformer and CNN to improve the model’s ability to segment the lesion edge region.

## Figures and Tables

**Figure 1 brainsci-12-00797-f001:**
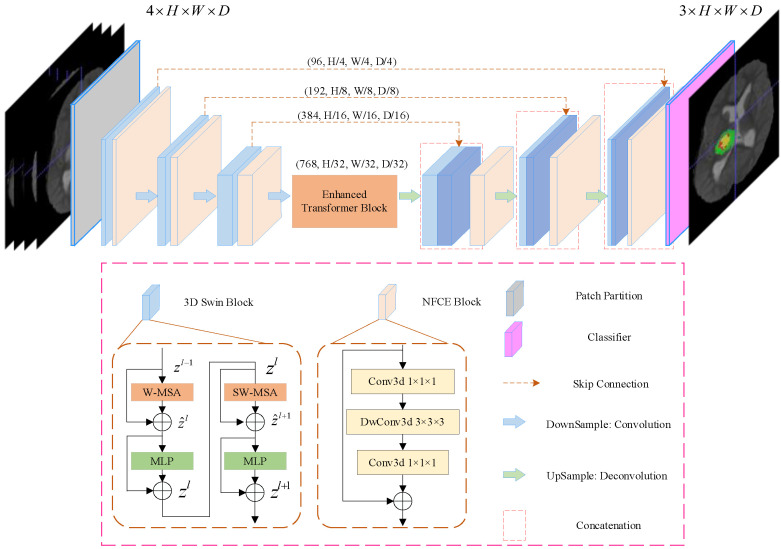
Overview of UNETR architecture. A 3D input volume (e.g., C=4 channels for MRI images) is divided into a sequence of uniform non-overlapping patches and projected into an embedding space using a linear layer. The sequence is added with a position embedding and used as an input to a transformer model. The encoded representations of different layers in the transformer are extracted and merged with a decoder via skip connections to predict the final segmentation.

**Figure 2 brainsci-12-00797-f002:**
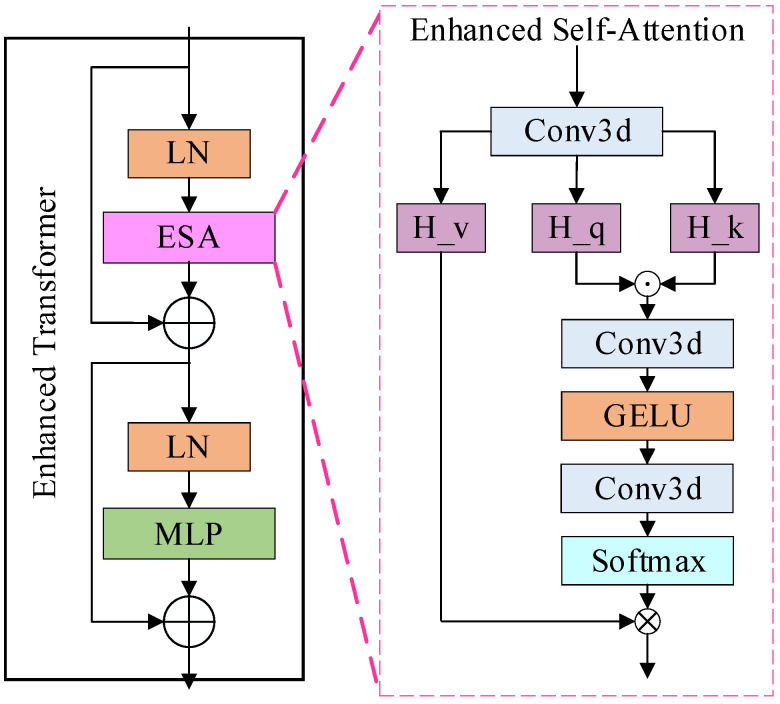
The structure of the enhanced transformer module.

**Figure 3 brainsci-12-00797-f003:**
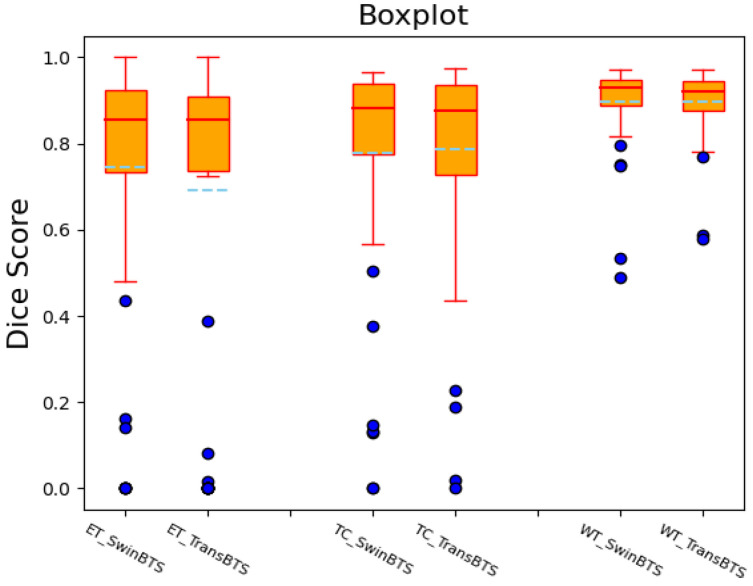
Dice score boxplot of SwinBTS and TransBTS methods on BraTS 2019 test dataset.

**Figure 4 brainsci-12-00797-f004:**
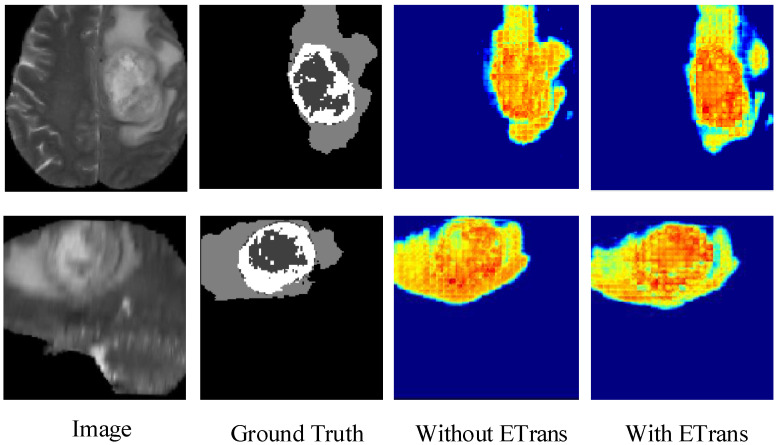
Comparing the effects of ETrans modules through heatmaps.

**Figure 5 brainsci-12-00797-f005:**
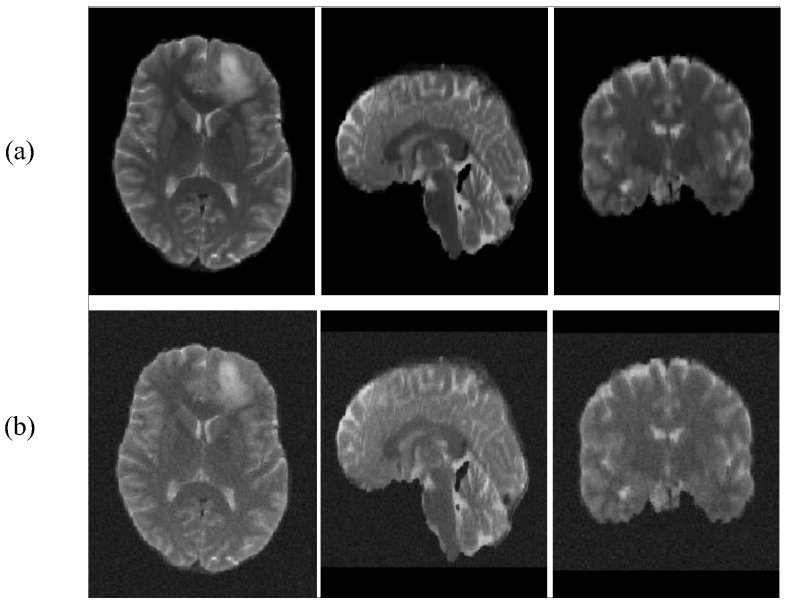
T2 images add noise contrast. (**a**): with no noise, (**b**): with noise.

**Figure 6 brainsci-12-00797-f006:**
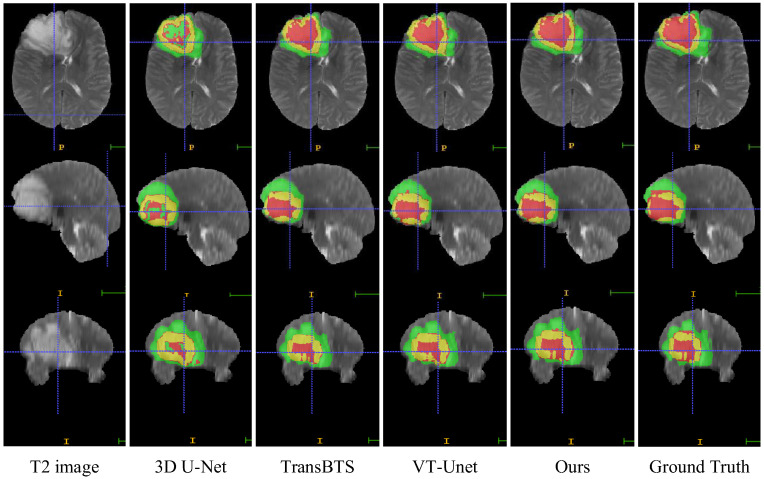
Visual comparison of MRI image segmentation results.

**Table 1 brainsci-12-00797-t001:** Segmentation Dice results on BraTS 2019 test set.

Method	Dice Score (%)			
	ET	TC	WT	AVG.
3D U-Net [31]	66.15 ± 0.339	66.94 ± 0.322	86.89 ± 0.071	73.33
Attention U-Net [48]	67.06 ± 0.327	71.95 ± 0.264	86.69 ± 0.100	75.23
U-Netr [27]	67.19 ± 0.346	74.39 ± 0.256	88.57 ± 0.122	76.72
TransBTS [38]	71.08 ± 0.347	78.67 ± 0.207	89.75 ± 0.070	79.83
VTU-Net [41]	73.53 ± 0.311	78.09 ± 0.242	89.56 ± 0.089	80.39
**SwinBTS**	74.43 ± 0.294	79.28 ± 0.232	89.75 ± 0.070	81.15

**Table 2 brainsci-12-00797-t002:** Segmentation mIOU results on BraTS 2019 test set.

Method	mIOU (%)			
	ET	TC	WT	AVG.
3D U-Net [31]	55.96 ± 0.308	52.72 ± 0.304	78.02 ± 0.104	73.33
Attention U-Net [48]	57.85 ± 0.309	61.73 ± 0.275	77.76 ± 0.137	75.23
TransBTS [38]	62.63 ± 0.322	69.16 ± 0.228	82.38 ± 0.100	79.83
VTU-Net [41]	65.00 ± 0.300	69.09 ± 0.251	81.12 ± 0.114	80.39
**SwinBTS**	66.03 ± 0.296	70.23 ± 0.216	83.33 ± 0.104	81.15

**Table 3 brainsci-12-00797-t003:** Segmentation results on the BraTS 2020 validation dataset.

Method	Dice Score (%)				95% Hausdorff Dist. (mm)			
	ET	TC	WT	AVG.	ET	TC	WT	AVG.
3D U-Net [31]	70.63 ± 0.284	73.70 ± 0.128	85.84 ± 0.250	76.72	34.30	18.86	10.93	21.36
V-Net [33]	68.97	77.90	86.11	77.66	43.52	16.15	14.49	24.72
Residual U-Net [49]	71.63	76.47	82.46	76.85	37.42	13.11	12.34	20.95
Attention U-Net [48]	71.83 ± 0.317	75.96 ± 0.126	85.57 ± 0.245	77.79	32.94	19.43	11.91	21.42
U-Netr [27]	71.18 ± 0.297	75.85 ± 0.100	88.30 ± 0.226	78.44	34.46	10.63	8.18	17.75
TransBTS [38]	76.31 ± 0.272	80.36 ± 0.075	88.78 ± 0.174	81.82	29.83	9.77	5.60	15.06
VTU-Net [41]	76.45 ± 0.267	80.39 ± 0.107	88.73 ± 0.218	81.86	28.99	14.76	9.54	17.76
SwinBTS	77.36 ± 0.224	80.30 ± 0.079	89.06 ± 0.130	82.24	26.84	15.78	8.56	17.06

**Table 4 brainsci-12-00797-t004:** Segmentation results on BraTS 2021 validation dataset.

Method	Dice Score (%)				95% Hausdorff Dist. (mm)			
	ET	TC	WT	AVG.	ET	TC	WT	AVG.
SwinBTS	83.21 ± 0.222	84.75 ± 0.227	91.83 ± 0.078	86.60	16.03	14.51	3.65	11.39

**Table 5 brainsci-12-00797-t005:** Ablation experiments of each module.

Model	Dice Score (%)			
	ET	TC	WT	AVG.
SwinUnet3D	71.75	76.74	88.40	78.96
SwinUnet3D + NFCE	73.00	77.48	89.01	79.83 (+0.87)
SwinUnet3D + NFCE + Trans	73.42	77.91	90.07	80.46 (+0.63)
SwinBTS + NFCE + Conv	72.55	77.97	87.85	79.46 (−0.37)
SwinUnet3D + NFCE + ETrans	74.43	79.28	89.75	81.15 (+1.32)

**Table 6 brainsci-12-00797-t006:** Experiments of different depths.

Method	Depth	Dice Score (%)			
		ET	TC	WT	AVG.
SwinBTS	1	73.06	78.60	89.08	80.24
	2	74.43	79.28	89.75	81.15
	4	72.88	79.19	89.63	80.57

**Table 7 brainsci-12-00797-t007:** Experiments of different degrees of noise.

Method	Noise-Sigma	Dice Score (%)			
		ET	TC	WT	AVG.
SwinBTS	0	74.43	79.28	89.75	81.15
	1	69.62	75.84	85.80	77.08
	5	59.67	69.85	82.85	70.79

## Data Availability

The datasets are provided by BraTS 2019 Challenge, BraTS 2020 Challenge, BraTS 2021 Challenge and are allowed for personal academic research. The specific link to the dataset is https://ipp.cbica.upenn.edu/ (accessed on 18 May 2022).

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
