# Peer review of "SwinBTS: A Method for 3D Multimodal Brain Tumor Segmentation Using Swin Transformer"

_brainsci, 2022, doi:10.3390/brainsci12060797_

Round 1

Reviewer 1 Report

In this study the authors propose SwinBTS, a new 3D medical picture segmentation approach to define the 3D brain tumor semantic segmentation job tested on extensive  BraTS 2019, BraTS 2020, and BraTS 2021 datasets. The results reveal that SwinBTS outperforms state-of-the-art 3D algorithms for brain tumor segmentation on 3D MRI scanned images. 
Statistical analyses have been conducted and figures and tables are informative and clear.
The paper is interesting, well written, I have some concern:
1)The link https://github.com/langwangdezhexue/SwinBTS doesn’t work
2)Could you estimate the differences in computation time between Transformer Model and Swin Transformer Model?
3)Probably the strengths of the article can be highlighted more in the conclusions. Any limitations? If there are none, you can write it in order to give more robustness to the method.

Reviewer 2 Report

The paper proposes a 3D medical image segmentation approach for brain tumor segmentation. The article has some methodological and presentation issues and needs to be revised according to the comments presented below.

Comments:

1.      Recently, many modifications and improvements of deep learning models were proposed and various applications were discussed. How does your model stands out? 

2.      The work also needs a very detailed presentation of how the values were found for the deep learning network hyperparameters (subsection 4.2), and how any overfitting was compensated for.

3.      Present more experimental results. Add the performance in terms of mean Intersection over Union.

4.      The improvement of the proposed method is small (Tables 1-2). Is it statistically significant? Perform the statistical analysis of the results. Use ANOVA, t-test or similar statistical tests.

5.      Why a similar table to tables 1 and 2 is not presented for the Brats 2021 dataset?

6.      Discuss your results from the viewpoint of explainability using, for example, Grad-CAM maps.

7.      The noise in real-world biomedical images is a well-known problem that reduces the accuracy of diagnostics. You should explore the robustness of the proposed method to noise in the X-ray images. You can artificially lower the quality for example by adding noise, and then analyze the drop in performance.

8.      Add the discussion section to discuss the limitations of this study. What are the practical implications of your study? I suggest to formulate recommendations for brain MRI segmentation by deep learning models.

9.      Improve conclusions. Use the main findings from the experiments to support your claims.

Round 2

Reviewer 2 Report

The authors are encouraged to compare the architecture they have used in this study with some of deep learning models and networks presented in Kadry, et al. (2021). Evaluation of brain tumor using brain MRI with modified-moth-flame algorithm and Kapur’s thresholding: A study; Maqsood, et al. (2021). An efficient approach for the detection of brain tumor using fuzzy logic and U-NET CNN classification; Muzammil, et al. (2020). CSID: A novel multimodal image fusion algorithm for enhanced clinical diagnosis. 

Author Response

Point 1: The authors are encouraged to compare the architecture they have used in this study with some of deep learning models and networks presented in Kadry, et al. (2021). Evaluation of brain tumor using brain MRI with modified-moth-flame algorithm and Kapur’s thresholding: A study; Maqsood, et al. (2021). An efficient approach for the detection of brain tumor using fuzzy logic and U-NET CNN classification; Muzammil, et al. (2020). CSID: A novel multimodal image fusion algorithm for enhanced clinical diagnosis.

Response 1: Thanks very much for the three papers provided by the reviewer. We have studied these papers carefully. One of these papers is about image fusion, and the rest of the segmentation methods used are more basic methods (threshold and UNET). Because the methods and datasets used are quite different, we cannot make a detailed comparison with our method, so we only cite it in our paper for introduction.

For details, see the introduction part of the first section of the paper, and we also highlight the modified parts.